# RECODE: Reasoning Through Code Generation for Visual Question Answering

## Abstract

Multimodal Large Language Models (MLLMs) struggle with precise reasoning for structured visuals like charts and diagrams, as pixel-based perception lacks a mechanism for verification. To address this, we propose to leverage *derendering*—the process of reverse-engineering visuals into executable code—as a new modality for verifiable visual reasoning. Specifically, we propose RECODE, an agentic framework that first generates multiple candidate programs to reproduce the input image. It then uses a critic to select the most faithful reconstruction and iteratively refines the code. This process not only transforms an ambiguous perceptual task into a verifiable, symbolic problem, but also enables precise calculations and logical inferences later on. On various visual reasoning benchmarks such as CharXiv, ChartQA, and Geometry3K, RECODE significantly outperforms methods that do not leverage code or only use code for drawing auxiliary lines or cropping. Our work shows that grounding visual perception in executable code provides a new path toward more accurate and verifiable multimodal reasoning.

## 1 Introduction

Multimodal large language models (MLLMs) have achieved remarkable progress in complex visual reasoning tasks, from providing detailed scene descriptions to understanding object relationships and temporal events (Gemini, 2025; OpenAI, 2025; Claude, 2025). However, there is still a critical gap in interpreting infographics, charts, and diagrams (Liu et al., 2022b; Masry et al., 2023; Wang et al., 2024; Tang et al., 2025; Masry et al., 2022). Unlike natural images, these visuals encode structured logic and quantitative relations. Accurately reasoning over them requires not only recognizing objects but also reconstructing the underlying generative logic. Current MLLMs, however, often attend to irrelevant regions, miss fine-grained details, or produce opaque reasoning chains with no mechanism to verify perceptual correctness (Masry et al., 2023; Huang et al., 2025b).

To improve reasoning, most existing approaches extend *language-based* reasoning pipelines to the visual domain (Shao et al., 2024; Chen et al., 2024; Rose et al., 2023). They extract visual descriptors like objects, attributes, and regions, translate them into natural language, and then attempt reasoning entirely in text. While effective for scene understanding, this paradigm is ill-suited for structured visuals: linguistic abstraction can discard crucial quantitative details, and the absence of an external check makes errors difficult to detect or correct. As a result, these methods struggle with the kind of *multi-step, verifiable reasoning* required for precise visual question answering for infographics.

We propose a fundamentally different approach that leverages *derendering*—the process of reverse-engineering visual inputs into executable code—as a new reasoning modality that simultaneously (1) provides a structured, interpretable representation of visual content, and (2) enables verification via re-rendering. Executed code generates an image that can be compared directly with the original visual, offering a concrete signal of perceptual fidelity. Moreover, once visuals are expressed in code, downstream reasoning can leverage computational tools for calculations, logical inference, and programmatic queries, which would be error-prone in natural language.

Prior efforts have leveraged code as a reasoning tool but treat it narrowly as either API calls to external vision models (Zheng et al., 2025; Huang et al., 2025a;b), or as limited "sketchpads" for drawing auxiliary lines or zooming-in and cropping (Hu et al., 2024; Zhang et al., 2025; Fan et al., 2025). These uses fall short of capturing the full generative logic of structured visuals. More im-

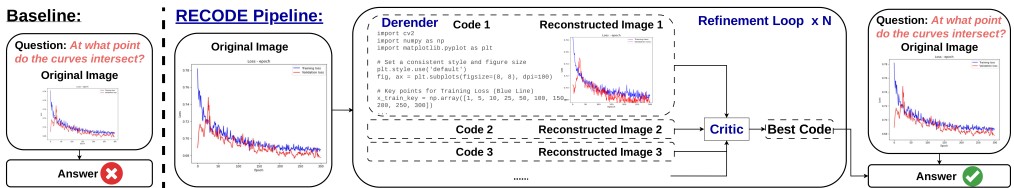

Figure 1: High-level architecture of our visual reasoning agent RECODE. Given an input image, the model first generates multiple candidate code programs. A critic function then selects the most faithful reproduction as the seed for next round of refinement. After the code is refined multiple times, the model uses both the generated code and the original image to answer the question.

portantly, they lack the potential of iterative self-correction. Our approach diverges fundamentally by positioning derendering not as a one-shot tool call, but as the core of *an agentic feedback loop*: the agent generates candidate code, executes it, critiques the reconstruction against the original, and iteratively refines the code until a faithful representation emerges.

Specifically, we propose **RECODE** (**RE**asoning via **CODE** generation) to improve visual reasoning and question solving (Figure 1). Given an input image, the RECODE agent first tries to generate code that reproduces it. Then, it engages in a closed-loop of iterative self-refinement. Inspired by the best-of-$n$ paradigm (Snell et al., 2024), the agent is prompted to explicitly identify discrepancies between its reconstruction and the original image, then autonomously revise its own code to minimize these errors over multiple cycles. To guide this refinement, the agent employs a critic-based selection mechanism. We benchmark multiple critics and find that the pixel-based Mean Squared Error (MSE) provides a robust and efficient signal for identifying the most faithful code representation among multiple candidates. To produce high-quality candidates for generation and refinement, we also develop a hierarchical derendering strategy, which decomposes the visual image into high-level and low-level components and integrates OCR for textual grounding.

We evaluate our agent on various visual reasoning benchmarks. On CharXiv-Reasoning (Wang et al., 2024), RECODE achieves 73% accuracy, a 15% gain over the baseline model that does not use de-rendering. This strong performance extends to the ChartQA dataset (Masry et al., 2022), where our agent achieves state-of-the-art performance of 93.2% accuracy, which is even 3% better than chart-pretrained foundation models like MatCha (Liu et al., 2022b). Besides charts, we also find benefits of derendering geometry diagrams and show strong results on formal diagrammatic reasoning with Geometry3K (Lu et al., 2021). These results provide strong evidence that derendering and iterative refinement can boost multimodal reasoning, offering both accuracy gains and interpretable, verifiable reasoning chains.

## 2 RELATED WORK

**Visual Reasoning Agents.** Large language models (LLMs) demonstrate enhanced reasoning abilities when augmented with external tools such as search engines, calculators, or Python interpreters (Schick et al., 2023; Qin et al., 2023; Liu et al., 2023). Programming-based methods in particular use code to decompose complex problems into executable steps, improving both transparency and accuracy, especially in mathematical reasoning (Gao et al., 2022; Chen et al., 2022). Inspired by this, recent multimodal approaches also start to leverage code generation to improve visual reasoning. For example, VisProg (Gupta & Kembhavi, 2022), ViperGPT (Sur'is et al., 2023), SAM R1 (Huang et al., 2025a) employ LLMs to generate Python code that sequentially invokes specialized vision models such as object detectors or segmenters. Beyond model invocation, methods like Visual Sketchpad (Hu et al., 2024) use code to draw intermediate visual artifacts that guide reasoning. Chain-of-focus (Zhang et al., 2025), VisualToolAgent (Huang et al., 2025b), DeepEyes (Zheng et al., 2025), and GRIT (Fan et al., 2025) focus more on cropping and zooming-in to a target region to help problem solving. While these works highlight the benefits of code, they use code as a tool for planning sequential actions rather than as a medium for deeply understanding the image itself. By contrast, our work converts visuals into executable generative programs, which enables models not only to externalize their perception in a verifiable form but also to iteratively refine it, going beyond the one-shot planning paradigm of prior tool-augmented agents.

**Benchmarks for Visual Reasoning.** Visual reasoning spans a spectrum from low-level perceptual tasks, such as depth estimation or object detection, to cognitively demanding domains like charts, diagrams, and infographics. The latter are particularly challenging because they require parsing heterogeneous elements (numbers, text, geometric structures) and performing precise quantitative reasoning. Recent benchmarks such as ChartQA (Masry et al., 2022), PlotQA (Methani et al., 2019), Charxiv (Wang et al., 2024), and ChartMuseum (Tang et al., 2025) focus on chart and infographic understanding, evaluating models on tasks like reading bar heights, comparing proportions, or extracting tabular information. Similarly, diagram-based benchmarks such as Geometry3K (Lu et al., 2021) and MathVista (Lu et al., 2023) test fine-grained diagram interpretation and text-conditioned reasoning. These settings expose the limitations of purely pixel-based perception pipelines and motivate approaches that can explicitly capture structure and support verifiable reasoning. Our method achieves strong performance across both chart and geometry benchmarks, demonstrating the effectiveness of adaptive, code-driven derendering for multimodal reasoning.

**Image Derendering.** The task of converting visual inputs back into structured, programmatic representations has a long history in computer vision, ranging from graphics program induction (Ellis et al., 2017) to chart-to-table conversion (Liu et al., 2022a;b). For instance, Matcha (Liu et al., 2022b) pre-trains models to derender charts into their underlying data tables. However, most prior work focuses narrowly on reconstruction accuracy rather than downstream reasoning, and does not integrate derendering into a self-improving agentic pipeline. Other works focus on chart comprehension with MLLMs (Masry et al., 2023; Fan et al., 2024). To our knowledge, RECODE is the first to leverage derendering for iterative self-refinement and systematically demonstrate its effectiveness as a core mechanism for improving downstream question answering on charts.

## 3 METHOD

To enable interpretable and verifiable reasoning over structured visuals, we propose to derender images into executable code that captures their underlying generative structure. This code serves both as a symbolic representation for reasoning and as a mechanism for self-verification: executing the code produces a reconstruction that can be directly compared to the input. In this section, we will first provide empirical evidence to motivate our derendering approach. Then, we will introduce the proposed RECODE pipeline that iteratively generates, critiques, and refines code representations before leveraging them for question answering (Figure 1).

### 3.1 PROOF-OF-CONCEPT: THE UTILITY OF CODE AS A REASONING MODALITY

**Definition of Derendering.** The generation of structured visuals, such as charts and diagrams, is fundamentally a programmatic process. An underlying script defines the rendering logic which governs layout, chart type, and aesthetics and integrates the source data. A rendering engine then interprets this specification to produce the final visual output. However, our approach is centered on the concept of *derendering*, which is commonly defined as the task of inferring this latent programmatic source from the input image alone (Liu et al., 2022b; Masry et al., 2023). When using a MLLM for derendering, we are forcing the model to form a deeper, symbolic understanding of the image, moving beyond pixel-level analysis to a structured representation.

Why is derendering potentially helpful for visual reasoning? We notice that pixel-based reasoning lacks a robust mechanism for verification; if a model misreads a single value from a chart, the entire reasoning chain can fail without a clear path to self-correction. We hypothesize that derendering can transform an ambiguous perceptual task into a more structured, symbolic reasoning problem. In the following, we first seek to validate this hypothesis and answer a fundamental question: *does access to the underlying generative code of an image fundamentally improve a model's visual reasoning capabilities?* Specifically, we design a proof-of-concept experiment to isolate and quantify the benefit of using code as a supplementary modality for visual question answering (VQA).

**Dataset Curation & Experiment Setup.** To obtain paired code and image examples, we curated a specialized dataset from the CharXiv-Reasoning dataset (Wang et al., 2024). Since CharXiv itself does not have ground-truth generative code, we sampled 200 examples and performed the following:

1. Code Generation: For each chart image, we prompted a state-of-the-art MLLM to generate Python code that reconstruct the input image.

2. Image Rendering: We executed this generated code to render a new, clean version of the chart.

3. Answer Verification and Correction: We manually reviewed each generated question-answer pair against the newly rendered image. Any answers that were no longer correct due to discrepancies in the rendered data were corrected. This verification step ensures a perfect alignment between the image, its code, and the ground-truth answer.

We use Gemini 2.5 Pro (Gemini, 2025) as our MLLM due to its remarkable multimodal performance among frontier models. The resulting dataset, which we name CharXiv-Mini, allows us to measure the impact of each technique in our method later. For proof-of-concept, we evaluate Gemini's performance under three settings: (1) Image-Only: The model is given only the original image and the question, i.e., the standard MLLM approach to visual reasoning; (2) Code-Only: The model is given only the generative Python code and the question, forcing it to reason over the symbolic and numerical data within the code. (3) Image + Code: The model receives the image and the code, allowing it to leverage information from both modalities. Detailed prompts are in Appendix A.1.1.

**Takeaway: Using Code Significantly Improves Visual Reasoning.** The results are summarized in Table 1. The Image-Only baseline achieved 75% accuracy, confirming the capabilities of modern MLLMs but also highlighting a substantial room for improvement. The Code-Only setting shows an 18% improvement over the baseline, demonstrating that access to generative code provides a substantial advantage for complex visual reasoning. The accuracy gap underscores that for questions requiring precise numerical extraction and logical inference, reasoning over a symbolic, machine-readable format is far more robust than reasoning over pixels. For instance, in an example asking which method's median is farthest from a reference line, the image-only model struggled, stating, "*It is difficult to determine a clear farthest method due to visual proximity.*" But the code-only model could deterministically identify the data generation process (`np.random.lognormal`) and calculate the exact theoretical median (`exp(mean)`) for each method, leading to a confident and correct answer. The Image + Code setting yielded the highest accuracy at 94%, suggesting that while the code contains most of the necessary information, the image can still provide useful grounding. The fact that this setting did not reach 100% is due to factors like incorrect understanding of questions or randomness in the code.

Table 1: QA Accuracy across different modality settings on our curated **CharXiv-Mini** dataset.

| Method | QA Accuracy |
|---|---|
| Image-Only | 75% |
| Code-Only | 93% |
| Image + Code | 94% |

These results provide strong motivation for our work. However, we note that in this experiment, the model is provided with the ground-truth code, but a practical agent must be able to produce this code itself. It is thus important to develop a stronger visual reasoning agent centered around high-fidelity derendering (code generation) and iterative refinement to improve code quality. In the next section, we introduce our agent designed to address these challenges.

### 3.2 AGENTIC PIPELINE

Given an input image $I$ and a natural language query $q$, the goal of a visual reasoning agent is to output an answer $a$. Unlike conventional prompting methods that rely solely on pixels and text, we introduce a structured intermediate representation $C$, an executable program (e.g., Python with Matplotlib) that encodes the content and layout of $I$. Our agent outputs a refined program $\hat{C}$ and a final answer $\hat{a}$. The correctness of code generation can be verified by comparing the rendering $R(\hat{C})$ with $I$, where $R$ denotes the rendering process. We design the agentic pipeline as follows (Figure 1):

1. **Multi-Candidate Code Generation**: Based on $I$, the agent generates multiple candidate programs $\{C_1, \ldots, C_k\}$ that plausibly reproduce $I$, leveraging the best-of-$n$ paradigm (Snell et al., 2024). To improve quality, we utilize an OCR tool and a hierarchical task decomposition process.

2. **Candidate Selection via Critic**: Each $C_i$ is executed to render $R(C_i)$. A critic evaluates similarity to $I$ and the top candidate $C^*$ is selected. The code corresponding to the highest-scoring rendered image is selected as the current best representation of the agent's understanding.

3. **Iterative Self-Refinement**: Starting from $C^*$, the agent enters a refinement loop. It is prompted with the original image, the best-so-far code, and the re-rendered image. The prompt explicitly asks it to identify discrepancies between the original and the reconstruction $\Delta(I, R(C^*))$ and

Table 2: Ablation study on code generation techniques, evaluated on **CharXiv-Mini**. All three proposed techniques improve the final VQA performance.

| Method | + Task Decomp | + Determinism | + OCR | CharXiv-Mini Accuracy |
|---|---|---|---|---|
| Image-Only (Baseline) | – | – | – | 75% |
| Derendering (Ablations) | – | – | – | 78% |
| | ✓ | – | – | 82% |
| | ✓ | ✓ | – | 85% |
| | ✓ | ✓ | pytesseract | 87% |
| | ✓ | ✓ | Gemini 2.5 Pro | 89% |
| Image + GT Code (Upperbound) | - | - | - | 94% |

then revise the code. After $T$ iterations, this yields a faithful code representation $\hat{C}$. This loop can be repeated multiple times, with each iteration producing a higher-fidelity code representation.

4. **Answer Question**: After the refinement rounds, the agent uses the final, high-fidelity code along with the original image to answer the question $q$.

### 3.2.1 STEP 1: MULTI-CANDIDATE CODE GENERATION

The pipeline begins with derendering: the agent generates multiple candidate programs $\{C_1, C_2, \ldots, C_k\}$ that represent plausible hypotheses of how $I$ was constructed. Unlike previous work that directly prompts the MLLMs to generate code, we improve code quality via two strategies.

**Task Decomposition.** Inspired by step-by-step reasoning in Chain-of-Thought (Wei et al., 2022), we decompose the complex task of derendering into a structured, hierarchical process. Our approach employs a two-level, coarse-to-fine decomposition (prompt is provided in Appendix A.2.1). First, we instruct the model to identify the number and layout of subplots within the figure (*subplot-level decomposition*). It then generates the code for each subplot independently before integrating them into a final, composite script. Next, within each subplot, the model further breaks down the task by programming individual visual components sequentially (*component-level decomposition*). This process typically addresses structural elements (e.g., axes, labels, titles) first, followed by the data representations themselves (e.g., bars, lines, scatter points). We conducted ablation studies using CharXiv-Mini. Table 2 shows that compared to a single-shot derendering baseline, the decomposition strategies boost performance.

**Determinism.** In our preliminary experiments, we found that the LLM often generates stochastic code that reproduces a chart's overall distribution (e.g., a Gaussian curve) instead of its exact data points. Because this code produces a different visualization on each run, it is unsuitable for precise quantitative questions. We therefore introduced a determinism constraint, explicitly prompting the model to avoid random functions and instead hard-code the observed data values.

**OCR Integration.** Textual information, such as data labels, axis titles, and units, is critical for accurately interpreting charts and geometric diagrams. To ensure this information is faithfully captured, we integrate an Optical Character Recognition (OCR) step into our pipeline prior to code generation. We first employ an OCR tool (e.g., the Pytesseract library or Gemini) to extract all text from the input image. This extracted text is then explicitly provided as additional context within the derendering prompt. See Appendix A.2.2 for more implementation details. As confirmed by our ablation study (Table 2), this OCR-enhanced approach significantly improves both the fidelity of the generated code and the accuracy on downstream reasoning tasks.

In summary, the results in Table 2 show that combining hierarchical decomposition, OCR integration, and determinism constraints leads to 89% accuracy on CharXiv-Mini, closing much of the gap to the 94% accuracy achieved using the ground-truth code itself. The remaining 5% performance gap stems from errors such as misinterpreting complex visual styles (e.g., a specific dash pattern in a line plot) that are challenging to resolve in a single pass. As we will demonstrate in Section 3.2.3, this remaining gap is precisely what our iterative self-refinement mechanism is designed to address.

### 3.2.2 STEP 2: CANDIDATE SELECTION VIA CRITIC

A key to our agent is the ability to assess the fidelity of the generated code and its corresponding rendered image. Given that multiple code-image pairs may be produced, a reliable "critic" is needed

Table 3: Visual question answering (VQA) accuracy using different critic functions to select the best candidate code-image pair out of five generated candidates. The number of Gemini and Gemini Embedding (Lee et al., 2025) calls required by each approach is also shown.

| Critic Function | # MLLM Calls | CharXiv-Mini Accuracy |
|---|---|---|
| *Pixel-Based Metrics* | | |
| EMD | 5+1 | 89% |
| MSE | 5+1 | **92%** |
| SSIM | 5+1 | 90% |
| PSNR | 5+1 | **92%** |
| *Embedding-Based Metric* | | |
| Embedding L2 | 5+5+1 | 88% |
| Embedding Cosine | 5+5+1 | 89% |
| *LLM-as-a-Judge* | | |
| Pairwise Assessment | 5+5+1 | 91% |
| Comparative Assessment | 5+1+1 | **92%** |

to select the most faithful representation, penalizing discrepancies in visual style as well as semantic errors that affect the data representation. We explore three classes of critic functions (implementation details and prompts can be found in Appendix A.3.1):

- **Pixel-Based Metrics** directly compare the pixel values of two images. We consider the Mean Squared Error (MSE), Structural Similarity Index Measure (SSIM), Peak Signal-to-Noise Ratio (PSNR), and Earth Mover's Distance (EMD). These metrics are computationally efficient but might be sensitive to small pixel-level differences that may not affect the chart's overall meaning.

- **Embedding-Based Metrics** leverage the semantic understanding of a pretrained model. We leverage Gemini Embedding (Lee et al., 2025) to obtain the image embeddings of the original and rendered charts and compute the L2 distance. This approach aims to capture high-level semantic similarity but can be less sensitive to fine-grained details.

- **LLM-as-a-Judge** frames the fidelity assessment as a MLLM reasoning task. Specifically, we prompt Gemini 2.5 Pro with both the original and generated images, along with a detailed rubric outlining the criteria for a good derendering. We consider two variations. (1) Pairwise: The model directly compares the original and rendered images, producing a single score reflecting the overall quality of the rendered image. (2) Comparative: The model evaluates all candidate images simultaneously, providing a ranking across the set of generated outputs.

**Takeaway: MSE Strikes a Balance Between Efficiency and Effectiveness.** We evaluated the effectiveness of each critic function by measuring its correlation with downstream QA performance on CharXiv-Mini. We set the number of candidates generated each round to 5. As shown in Table 3, among all metrics, MSE and PSNR all achieved the best performance, despite being simple and computationally efficient. The embedding-based metrics performed slightly worse, suggesting that they are less sensitive to the detailed visual features required for accurate chart understanding. While LLM-as-a-judge-comparative also achieves good performance, the empirical cost associated with calling APIs makes it less appealing compared to simply computing the MSE. Given its simplicity and performance, we select the MSE metric as the critic for future stages of our RECODE agent.

### 3.2.3  STEP 3: ITERATIVE SELF-REFINEMENT

The initial derendering process provides a strong foundation for reasoning, but a single-pass generation may not capture all the nuances of a complex visual. To address this, we introduce an iterative self-refinement mechanism, enabling the agent to progressively improve its own generated code by comparing its rendered output with the original image and correcting discrepancies.

The refinement process operates as a loop that takes the best code from the previous step as a seed for the next round of generation. Formally, given the current best candidate $C^*$, the agent is explicitly asked to: (1) analyze the original image $I$, (2) analyze the reconstructed images $R(C^*)$, and (3) identify discrepancies between $R(C^*)$ and $I$ (e.g., misaligned labels, incorrect bar heights). It then produces a revised program $C'$, which is re-scored by the critic. This process repeats for $T$

Table 4: Performance improvement for different refinement rounds. Each round improves the visual fidelity of the derendered image (lower MSE) and boosts the downstream QA accuracy.

| Refinement Round | MSE (original, generated) | CharXiv-Mini Accuracy |
|---|---|---|
| 0 (No Refinement) | 2325 | 92% |
| 1 | 2030 | 94% |
| 2 | 1913 | 95% |

refinement rounds, yielding progressively more faithful programs $\hat{C}$:

$$C^{(t+1)} = \text{Refine}(C^{(t)}, \Delta(I, R(C^{(t)}))),$$

where $\Delta(\cdot)$ denotes detected discrepancies. This iterative correction is critical to capture fine-grained details and correct errors that one-shot code generation cannot resolve. For instance, if it initially misreads a data point at $y = 75$ as $y = 72$, the discrepancy in the re-rendered image will be salient, prompting a correction in the next refinement iteration.

Note that the refinement process is also executed to generate *multiple* new code candidates in parallel. The critic is then used to select the best candidate from this new set, which becomes the seed for the subsequent refinement round. The refinement prompt is provided in Appendix A.4.1.

To evaluate the effectiveness of our approach, we conducted experiments over multiple refinement rounds on CharXiv-Mini. In each round, the agent generated five new candidate codes, from which the single best code-image pair was selected using the MSE critic. We tracked two key metrics: the MSE between the original and generated images to quantify improvements in visual fidelity, and the downstream QA accuracy to measure the impact on the agent's reasoning capabilities.

Table 4 demonstrates a consistent improvement across both visual fidelity and reasoning accuracy with each round of refinement. This supports our hypothesis that by forcing the agent to produce a more accurate representation of the visual, we enhance its underlying understanding, which in turn leads to more reliable reasoning. The gains show a pattern of diminishing returns, which is expected as the code becomes progressively more accurate and the remaining errors become subtler. Notably, after two rounds of refinement, our agent achieves 95% accuracy. This not only closes the gap to the ground-truth code performance (94%) but slightly surpasses it. The 1% improvement is due to the fact that the initial ground-truth code may be overly complex, whereas our agent generates a simplified version, which eases the difficulty of reasoning.

## 4 EXPERIMENTS

Having detailed the pipeline of our derendering agent and the design choices that informed its development, we now systematically evaluate it across various chart and geometry diagram benchmarks.

### 4.1 EXPERIMENTAL SETUP

All experiments are conducted using Gemini 2.5 Pro as the core model for both our agent and relevant baselines, ensuring a fair comparison of reasoning capabilities. Unless otherwise specified, we use two rounds of iterative refinement, generating five candidates per round. The best candidate is selected using the MSE critic. Detailed prompts for all tasks and agent configurations can be found in Appendix A.5. For all VQA tasks, we report *accuracy*, following the standard evaluation protocols of each respective dataset. We compare our method against the following baselines:

- Direct Prompting: We evaluate the direct reasoning capability of the underlying model by providing frontier models like Gemini (Gemini, 2025), Claude (Claude, 2025) with only the input image and the question, without any of the agentic scaffolding, code generation, or iterative refinement from our proposed method.

- Visual Reasoning Agents and Specialized Baselines: We compare against state-of-the-art visual reasoning agents such as VisualToolAgent (Huang et al., 2025b), and Visual Sketchpad (Hu et al., 2024), as well as specialized chart-pretrained foundation models such as MatCha (Liu et al., 2022b) and CoSyn (Yang et al., 2025).

Table 5: Main results on the **CharXiv-Reasoning** dataset, showing the impact of our proposed method with iterative refinement. Our method uses 5 candidates per round.

| Method | Uses Image | Uses Code | # Refinement Rounds | QA Accuracy |
|---|---|---|---|---|
| Human | $\checkmark$ | $\times$ | - | 80% |
| *Direct Prompting* | | | | |
| Claude 3.7 | $\checkmark$ | $\times$ | - | 64% |
| GPT 4.5 | $\checkmark$ | $\times$ | - | 55% |
| Gemini 2.5 Pro | $\checkmark$ | $\times$ | - | 58% |
| **RECODE (Ours)** | $\checkmark$ | $\checkmark$ | 0 | 73% |
| | $\checkmark$ | $\checkmark$ | 1 | 76% |
| | $\checkmark$ | $\checkmark$ | 2 | 77% |

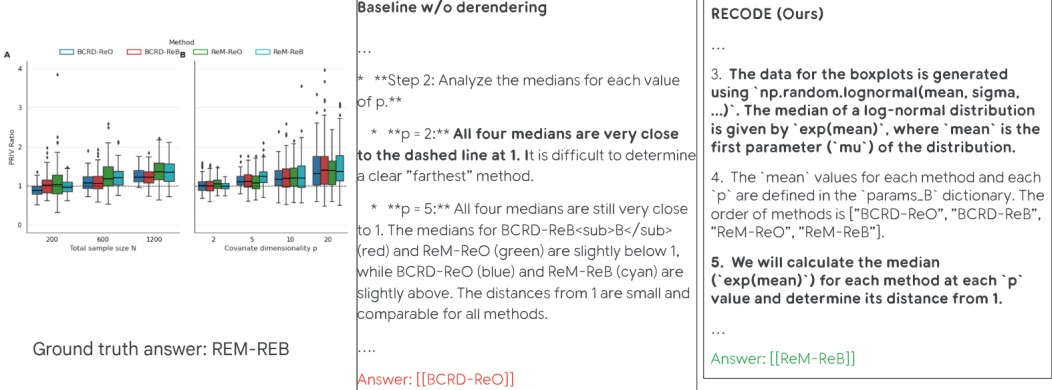

*Which method in plot B consistently shows medians farthest to the dashed horizontal line at PRV ratio = most frequently across all covariate dimensionalities p?*

Figure 2: A case study with comparison of reasoning traces. Baseline Gemini struggles, whereas our agent uses code to produce a confident and correct answer.

## 4.2 Evaluation on Scientific Charts: CharXiv-Reasoning

We first evaluate on the full CharXiv reasoning benchmark (Wang et al., 2024), which contains 1,000 VQA examples featuring complex scientific charts from academic papers. Our main results are summarized in Table 5. The direct prompting baseline achieves an accuracy of 58.0%. Our agent, even without any refinement rounds, already provides a significant boost to 73%. This demonstrates the immediate benefit of converting the image to code.

Most importantly, the iterative refinement process yields consistent improvements. After one round of refinement, accuracy increases to 76%, and after two rounds, it reaches 77%. This is a 19% absolute improvement over the baseline, highlighting the effectiveness of our agent's ability to self-correct its visual perception.

To provide a concrete example of our agent's reasoning process, we present a case study in Figure 2. The task is to identify which method in a boxplot "consistently shows medians farthest to the dashed horizontal line at PRV ratio = 1". This question is challenging for pixel-based models due to the visual proximity of the median lines, making direct perception unreliable. As shown in the baseline panel, the direct prompting baseline struggles, incorrectly concluding that one method is slightly farther when, visually, they are nearly indistinguishable. This perceptual ambiguity leads to a wrong answer. In contrast, our agent derenders the chart into its underlying generative code, which reveals the mean parameter (mu) for each method. By analyzing this code, our agent deduces the theoretical median for all four methods and answers the question correctly.

## 4.3 Robustness on Additional Benchmark: ChartQA

Having demonstrated strong performance on CharXiv, we further assess our agent's robustness on the ChartQA benchmark (Masry et al., 2022) which contains charts authored by humans for a wide

Table 6: Accuracy on the **ChartQA** test set.

| Method | Accuracy (%) |
|---|---|
| CoSyn-7B (Yang et al., 2025) | 86.3 |
| UniChart (Masry et al., 2023) | 88.6 |
| MatCha (Liu et al., 2022b) | 90.2 |
| Gemini 2.5 Pro | 89.4 |
| **RECODE (Ours)** | **93.2** |

Table 7: Accuracy on the **Geometry3K** test set.

| Method | Accuracy (%) |
|---|---|
| VisTA-QwenVL 7B (Huang et al., 2025b) | 55.6 |
| Inter-GPS (Lu et al., 2021) | 57.5 |
| Visual Sketchpad GPT-4o (Hu et al., 2024) | 66.7 |
| Gemini 2.5 Pro | 90.8 |
| **RECODE (Ours)** | **94.2** |

variety of topics, accompanied by complex, free-form questions. We evaluate RECODE on the test split of ChartQA.

As shown in Table 6, RECODE achieves the best result over all baselines. This confirms that our agent's structured reasoning process is highly effective on a large and diverse corpus of human-created charts. We observe that a common failure mode for the baseline is confusing data series in a multi-line graph when colors are similar or lines intersect frequently. In contrast, RECODE can detect when its rendered chart mismatches the original's legend or a specific data point. It then corrects its code to accurately map each legend entry to the correct line data. This ability to disentangle and verify complex visual information is key to its performance, demonstrating its practical utility for real-world chart understanding.

### 4.4 EXTENDING TO MATHEMATICAL REASONING: GEOMETRY3K

To test the applicability of our derendering framework beyond charts in research papers, we evaluate its performance on formal geometric reasoning using the Geometry3K benchmark (Lu et al., 2021). This dataset is composed of high-school level geometry problems, where each example includes a diagram, textual premises, and a question that requires multi-step logical deduction. This domain presents a unique challenge: success depends not only on extracting plotted data, but also on correctly identifying geometric entities (e.g., points, lines, circles), their properties (e.g., lengths, angles), and their relationships (e.g., perpendicularity, tangency). For this task, we ask our agent to generate Python code using matplotlib, networkx, and SymPy to reconstruct the geometric diagram before answering the question.

As shown in Table 7, RECODE demonstrates strong performance, significantly outperforming the direct prompting baseline. This highlights the power of converting an implicit visual diagram into an explicit, symbolic representation. The baseline Gemini can often make perceptual errors, such as misinterpreting an angle as 90 degrees or failing to correctly identify points of tangency. By forcing the agent to generate formal geometric code, RECODE mitigates these errors. The code serves as a structured "scratchpad" where all entities and their properties are explicitly defined, allowing the agent to perform multi-step deductions with a computational solver, which is far more reliable than attempting to reason holistically over pixels and text.

For example, when asked to find the length of a segment in a complex diagram involving circles and triangles, the baseline might fail to apply the Pythagorean theorem correctly because it misidentifies the right angle. RECODE, in contrast, would first derender the diagram into code that explicitly declares 'Triangle(A, B, C)' and 'is_right_angle(A, B, C)'. This symbolic grounding makes the subsequent application of the theorem trivial.

## 5 CONCLUSION

In this work, we introduced a visual reasoning agent that leverages derendering and iterative refinement to achieve a more robust and accurate understanding of chart-based images. By converting ambiguous pixel information into precise, executable code, our agent can verify its own perception and perform complex reasoning. The proposed iterative loop of generation, critic-based selection, and refinement allows the agent to progressively correct errors. Our experiments demonstrate a substantial improvement in QA accuracy on challenging visual reasoning benchmarks. A promising future direction is to collect agent trajectories for reinforcement learning.

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

# A APPENDIX

## A.1 PROOF-OF-CONCEPT EXPERIMENT DETAILS

### A.1.1 GEMINI PROMPTS

---

**Image-Only Prompt**

Based on the input image, answer the question: {question} In your response, first think step-by-step and reason about the question. Provide evidence for any reasoning. Then, output your answer in the format of: "Answer: [[...]]" Do not use markdown format or output anything else after "Answer".

---

**Code-Only Prompt**

Based on the Python code of a diagram image, answer the question: {question} In your response, first think step-by-step and reason about the question. Provide evidence for any reasoning. Then, output your answer in the format of: "Answer: [[...]]" Do not use markdown format or output anything else after "Answer".
The code is: {code}

---

**Image + Code Prompt**

Based on an input image and the Python code that generates this image, answer the question: {question} In your response, first think step-by-step and reason about the question. Provide evidence for any reasoning. Then, output your answer in the format of: "Answer: [[...]]" Do not use markdown format or output anything else after "Answer".
The code is: {code}

---

## A.2 DERENDERING

### A.2.1 TASK DECOMPOSITION PROMPTS

---

**Task Decomp Prompt**

You are an expert in Python for data visualization. Your specialty is reverse-engineering charts and diagrams from images into clean, reproducible code.
Your goal is to carefully analyze the provided chart or diagram and write Python code to generate a high-fidelity replica. Your response must follow the structure below.
Step 1: Identify Subfigures Identify how many subfigures are in the chart. Then, for each subfigure, repeat step 2 and step 3.
Step 2: Analysis and Data Extraction Provide a structured analysis of the chart. This is your plan for the code.
- Chart Type: Identify the primary type of chart (e.g., bar chart, line plot, scatter plot, pie chart, flowchart, schematic).
- Styling & Structure: Detail the visual style. List all structural elements (titles, labels, legends, annotations), colors, fonts, and line styles. Make sure the order of methods/legends of the chart is preserved.
- Data Inference: This is the most critical step. Infer the approximate data and relationships from the visual elements.
- For bar/line/scatter plots, estimate the data points and describe the axes (range, ticks, labels).
- For pie charts, estimate the percentage for each slice.
- For flowcharts/schematics, describe the nodes (shapes, text) and the connections between them (arrows, lines).
- For other chart types, estimate the data points and describe the axes (range, ticks, labels).

---

- Pay attention to text information available in the chart.
Step 3: Code Generation Next, write a chunk of Python code to generate the diagram based on your analysis. Make sure the chart type, structural elements, and the exact data points are preserved.
Other requirements and constraints:
- You must only use the following libraries: cv2, numpy (as np), matplotlib.pyplot (as plt), math, and seaborn (as sns).
- Do not define functions or classes. No need to define a main function as well. Just write the code block as if you are in a colab environment.
- The code should be self-contained for generation. Do not include image display code (e.g., plt.show(), cv2.imshow()) or package installation commands (e.g., !pip install).
- Do not use modules that involves randomness, such as np.random.
Step 4: Putting Together Now, chain all the code chunks together into a single chunk, which users can directly execute to get the full diagram. However, note that the final generated image must be a NumPy array named 'image_cv2' in BGR color format (the standard for OpenCV). To convert a matplotlib figure to the required format, you will need to:
- Draw the plot to the figure's canvas.
- Render the canvas to an RGBA NumPy array.
- Convert the RGBA array to a BGR array using cv2.cvtColor.
In your final output, make sure that:
- The entire code block is enclosed within "'python ..."
- Every variable is defined before being referred to.
- Do not define helper functions.
- The final image must be stored in a variable named 'image_cv2'.

### A.2.2 OCR Implementation

**Gemini Prompt**

You will be given an input image that's a chart or diagram. Carefully read the image and extract all text components from the image, including the title, label, data values, etc. Your should describe the text content, the position, just like you are describing the image to someone who does not have access to the image. Output a single paragraph.

For `pytesseract`, we implement as follows:

```
gray = cv2.cvtColor(original_image, cv2.COLOR_BGR2GRAY)

#Apply a binary threshold.
# Otsu's thresholding automatically determines the best threshold value.
# THRESH_BINARY_INV makes the text white and the background black, which can help Tesseract.
_, thresh_img = cv2.threshold(gray, 0, 255, cv2.THRESH_BINARY_INV + cv2.THRESH_OTSU)
custom_config = r'--oem 3 --psm 6'
text = pytesseract.image_to_string(thresh_img, config=custom_config)
```

## A.3 Critic

### A.3.1 LLM-as-a-Judge Prompts

**Autorater Prompt**

You are an expert in scientific diagram understanding. Your goal is to rate an AI-generated diagram against an original source diagram from a scientific paper. You will determine if the generated image is a faithful and accurate reproduction. You will be given the following inputs:
original_image: The ground truth diagram.
generated_image: The generated diagram by an AI model.

Your first task is to describe what's in each image. Then, analyze the generated image in comparison to the original image. Use the qualitative scale (excellent/good/fair/bad/terrible) for the quality assessments. You should only focus on the semantic accuracy, so stylistic differences such as color, boundary, line thickness, etc, can be safely ignored. Format your judgment exactly in the following way, do not add any extra symbols:

Analysis - Semantic Fidelity to Original: excellent/good/fair/bad/terrible

Analysis - Text & Label Accuracy: excellent/good/fair/bad/terrible

Analysis - Data Accuracy: excellent/good/fair/bad/terrible

Analysis - Artifacts & Hallucinations: none/minor/some/many/lots

After the rubric analysis, average the scores (excellent/none=5, good/minor=4, fair/some=3, bad/many=2, terrible/lots=1). Then, express your final judgment in the format: "Final verdict: [[score]]" where the score is the calculated average. Do not use markdown format or output anything else after the final verdict.

## A.4 REFINEMENT

### A.4.1 PROMPTS

**Refinement Prompt**

Your task is to reconstruct the given diagram by modifying the following code: {code}

You will be given the original diagram. Do the following:

1. Describe what's in the original input image. If there are multiple subfigures, describe each of them.

2. Describe what's in the reconstructed image from the given code. If there are multiple subfigures, describe each of them.

3. Identify the discrepancies between the original image and the reconstruction code. If there are multiple subfigures, do this for each of them. Pay attention to the semantic information (chart types, data points, etc) and the visual style (colors, titles, labels, legends, etc).

4. Revise the code to remove as many discrepancies as possible so that the new code faithfully reconstruct the original image. Note that the final generated image must be a NumPy array named 'image_cv2' in BGR color format (the standard for OpenCV). To convert a matplotlib figure to the required format, you will need to:

- Draw the plot to the figure's canvas.

- Render the canvas to an RGBA NumPy array.

- Convert the RGBA array to a BGR array using cv2.cvtColor.

In your final output, enclose the entire refined code block within "'python ...'". Make sure that:

- You must only use the following libraries: cv2, numpy (as np), matplotlib.pyplot (as plt), math, and seaborn (as sns).

- Do not define functions or classes. No need to define a main function as well. Just write the code block as if you are in a colab environment.

- The code should be self-contained for generation. Do not include image display code (e.g., plt.show(), cv2.imshow()) or package installation commands (e.g., !pip install).

- Every variable is defined before being referred to. - Do not use modules that involves randomness, such as np.random.

- The final image must be stored in a variable named 'image_cv2'.

## A.5 EXPERIMENT DETAILS

## A.6 CASE STUDIES

### A.6.1 EXAMPLE 1

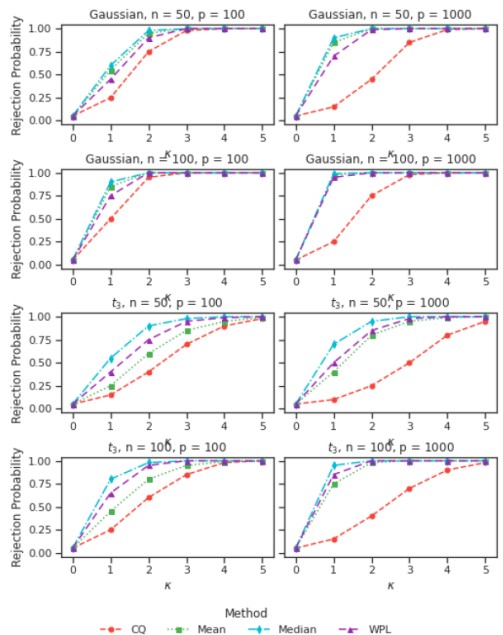

Figure 3: Original Image.

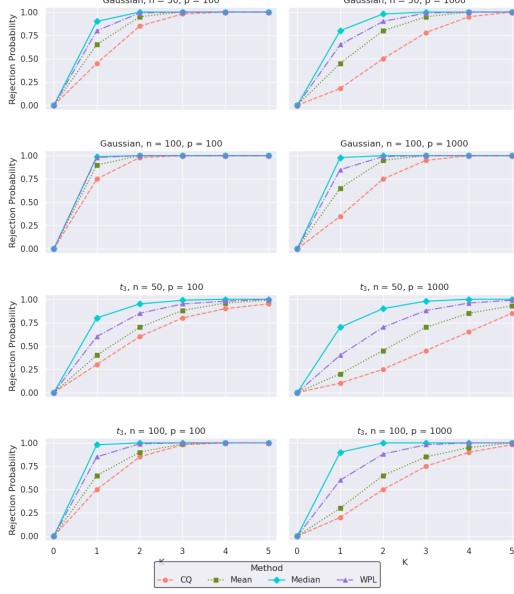

Figure 4: Derendered Image.

Generated code:

```python
import cv2
import numpy as np
import matplotlib.pyplot as plt
import seaborn as sns
import math
import matplotlib.ticker as mticker
```

```python
# --- Step 1 & 2: Data and Style Definition ---

# Set the visual style to match the original image
sns.set_theme(style="darkgrid")

# Define the data for all 8 subplots
plot_data = [
    {
        "title": "Gaussian, n = 50, p = 100",
        "data": {
            "CQ": [0.00, 0.45, 0.85, 0.98, 1.00, 1.00],
            "Mean": [0.00, 0.65, 0.95, 1.00, 1.00, 1.00],
            "Median": [0.00, 0.90, 1.00, 1.00, 1.00, 1.00],
            "WPL": [0.00, 0.80, 0.99, 1.00, 1.00, 1.00],
        }
    },
    {
        "title": "Gaussian, n = 50, p = 1000",
        "data": {
            "CQ": [0.00, 0.18, 0.50, 0.78, 0.95, 1.00],
            "Mean": [0.00, 0.45, 0.80, 0.95, 1.00, 1.00],
            "Median": [0.00, 0.80, 0.98, 1.00, 1.00, 1.00],
            "WPL": [0.00, 0.65, 0.90, 0.99, 1.00, 1.00],
        }
    },
    {
        "title": "Gaussian, n = 100, p = 100",
        "data": {
            "CQ": [0.00, 0.75, 0.98, 1.00, 1.00, 1.00],
            "Mean": [0.00, 0.90, 1.00, 1.00, 1.00, 1.00],
            "Median": [0.00, 0.99, 1.00, 1.00, 1.00, 1.00],
            "WPL": [0.00, 0.98, 1.00, 1.00, 1.00, 1.00],
        }
    },
    {
        "title": "Gaussian, n = 100, p = 1000",
        "data": {
            "CQ": [0.00, 0.35, 0.75, 0.95, 1.00, 1.00],
            "Mean": [0.00, 0.65, 0.95, 1.00, 1.00, 1.00],
            "Median": [0.00, 0.98, 1.00, 1.00, 1.00, 1.00],
            "WPL": [0.00, 0.85, 0.99, 1.00, 1.00, 1.00],
        }
    },
    {
        "title": r"$t_3$, n = 50, p = 100",
        "data": {
            "CQ": [0.00, 0.30, 0.60, 0.80, 0.90, 0.95],
            "Mean": [0.00, 0.40, 0.70, 0.88, 0.96, 0.99],
            "Median": [0.00, 0.80, 0.95, 0.99, 1.00, 1.00],
            "WPL": [0.00, 0.60, 0.85, 0.95, 0.98, 1.00],
        }
    },
    {
        "title": r"$t_3$, n = 50, p = 1000",
        "data": {
            "CQ": [0.00, 0.10, 0.25, 0.45, 0.65, 0.85],
            "Mean": [0.00, 0.20, 0.45, 0.70, 0.85, 0.93],
            "Median": [0.00, 0.70, 0.90, 0.98, 1.00, 1.00],
            "WPL": [0.00, 0.40, 0.70, 0.88, 0.96, 0.99],
        }
    },
    {
        "title": r"$t_3$, n = 100, p = 100",
        "data": {
            "CQ": [0.00, 0.50, 0.85, 0.98, 1.00, 1.00],
            "Mean": [0.00, 0.65, 0.90, 0.99, 1.00, 1.00],
            "Median": [0.00, 0.98, 1.00, 1.00, 1.00, 1.00],
            "WPL": [0.00, 0.85, 0.99, 1.00, 1.00, 1.00],
        }
    },
    {
        "title": r"$t_3$, n = 100, p = 1000",
        "data": {
            "CQ": [0.00, 0.20, 0.50, 0.75, 0.90, 0.98],
            "Mean": [0.00, 0.30, 0.65, 0.85, 0.95, 1.00],
            "Median": [0.00, 0.90, 1.00, 1.00, 1.00, 1.00],
            "WPL": [0.00, 0.60, 0.88, 0.98, 1.00, 1.00],
        }
    }
]
```

```
918     # Shared x-axis values
919     K = np.arange(6)
920
921     # Define styles for each method to ensure consistency and correct ordering
922     method_styles = {
            'CQ': {'color': '#FA8072', 'marker': 'o', 'linestyle': '--', 'label': 'CQ'},
923         'Mean': {'color': '#6B8E23', 'marker': 's', 'linestyle': ':', 'label': 'Mean'},
            'Median': {'color': '#00CED1', 'marker': 'D', 'linestyle': '-', 'label': 'Median'},
924         'WPL': {'color': '#9370DB', 'marker': '^', 'linestyle': '-.', 'label': 'WPL'}
        }
925
926     # --- Step 3: Code Generation ---
927
        # Create a 4x2 grid of subplots with a more accurate aspect ratio
928     fig, axes = plt.subplots(4, 2, figsize=(10, 12), sharex=True, sharey=True)
929
        # Iterate through each subplot's data and corresponding axis
930     for i, ax in enumerate(axes.flatten()):
            item = plot_data[i]
931         ax.set_title(item['title'])
932
            # Plot each method's data on the current axis
933         for method_name, style in method_styles.items():
                y_data = item['data'][method_name]
934             ax.plot(K, y_data, **style)
935
            # Configure axis limits and ticks
936         ax.set_ylim(-0.05, 1.05)
            ax.set_yticks(np.arange(0, 1.01, 0.25))
937         ax.set_xlim(-0.2, 5.2)
            ax.set_xticks(np.arange(0, 6, 1))
938
            # Format y-axis labels to two decimal places
940         ax.yaxis.set_major_formatter(mticker.FormatStrFormatter('%.2f'))
941
942     # Set shared axis labels for the entire figure
        for ax in axes[:, 0]:
943         ax.set_ylabel('Rejection_Probability')
        for ax in axes[-1, :]:
944         ax.set_xlabel('K')
945
        # Create a single, shared legend at the bottom of the figure
946     handles, labels = axes[0, 0].get_legend_handles_labels()
947     fig.legend(
            handles, labels,
948         title='Method',
            loc='lower_center',
949         bbox_to_anchor=(0.5, 0.04), # Adjusted legend position
            ncol=4,
950         frameon=True,
            edgecolor='black'
951     )
952
953     # Adjust subplot layout to prevent titles/labels from overlapping and to make space for the legend
954     plt.subplots_adjust(top=0.96, bottom=0.12, left=0.08, right=0.98, hspace=0.4, wspace=0.05)
955
956     # --- Final Conversion to OpenCV format ---
957
        # Draw the canvas
958     fig.canvas.draw()
959
        # Convert the matplotlib figure to an RGBA NumPy array
960     rgba_buf = fig.canvas.buffer_rgba()
        image_rgba = np.asarray(rgba_buf)
961
962     # Convert RGBA to BGR for OpenCV compatibility
        image_cv2 = cv2.cvtColor(image_rgba, cv2.COLOR_RGBA2BGR)
963
```

Question: How many subplots depict the Median and CQ methods reaching a rejection probability of 0.75 when K = 3.5?

### RECODE Solution

This question is more easily and accurately solvable by looking at the code. The question asks for the rejection probability at K = 3.5, which is not an explicit data point on the x-axis. To find the value at this point, we must interpolate between the given data points at K = 3

and K = 4. Visual interpolation from the image can be imprecise. The code provides the exact numerical values, allowing for a precise calculation.

The problem requires us to count the number of subplots where both the 'Median' method and the 'CQ' method have a rejection probability of at least 0.75 at K = 3.5. Assuming linear interpolation, the value at K = 3.5 is the average of the values at K = 3 and K = 4. The data for K values from 0 to 5 are stored in lists, so we need to look at the values at index 3 (for K=3) and index 4 (for K=4) for each method in each subplot.

Let's analyze each subplot's data from the 'all_data' list:

1. **Subplot 1:** * 'CQ': Values at K=3 and K=4 are 0.90 and 0.98. Interpolated value at K=3.5: (0.90 + 0.98) / 2 = 0.94 (= 0.75). * 'Median': Values at K=3 and K=4 are 1.0 and 1.0. Interpolated value at K=3.5: (1.0 + 1.0) / 2 = 1.0 (= 0.75). * Both conditions are met. (Count = 1)

2. **Subplot 2:** * 'CQ': Values at K=3 and K=4 are 0.75 and 0.95. Interpolated value at K=3.5: (0.75 + 0.95) / 2 = 0.85 (= 0.75). * 'Median': Values at K=3 and K=4 are 1.0 and 1.0. Interpolated value at K=3.5: (1.0 + 1.0) / 2 = 1.0 (= 0.75). * Both conditions are met. (Count = 2)

3. **Subplot 3:** * 'CQ': Values at K=3 and K=4 are 0.99 and 1.0. Interpolated value at K=3.5: (0.99 + 1.0) / 2 = 0.995 (= 0.75). * 'Median': Values at K=3 and K=4 are 1.0 and 1.0. Interpolated value at K=3.5: (1.0 + 1.0) / 2 = 1.0 (= 0.75). * Both conditions are met. (Count = 3)

4. **Subplot 4:** * 'CQ': Values at K=3 and K=4 are 0.95 and 1.0. Interpolated value at K=3.5: (0.95 + 1.0) / 2 = 0.975 (= 0.75). * 'Median': Values at K=3 and K=4 are 1.0 and 1.0. Interpolated value at K=3.5: (1.0 + 1.0) / 2 = 1.0 (= 0.75). * Both conditions are met. (Count = 4)

5. **Subplot 5:** * 'CQ': Values at K=3 and K=4 are 0.80 and 0.90. Interpolated value at K=3.5: (0.80 + 0.90) / 2 = 0.85 (= 0.75). * 'Median': Values at K=3 and K=4 are 0.99 and 1.0. Interpolated value at K=3.5: (0.99 + 1.0) / 2 = 0.995 (= 0.75). * Both conditions are met. (Count = 5)

6. **Subplot 6:** * 'CQ': Values at K=3 and K=4 are 0.55 and 0.75. Interpolated value at K=3.5: (0.55 + 0.75) / 2 = 0.65 ( 0.75). * The CQ condition is not met. (Count remains 5)

7. **Subplot 7:** * 'CQ': Values at K=3 and K=4 are 0.92 and 0.98. Interpolated value at K=3.5: (0.92 + 0.98) / 2 = 0.95 (= 0.75). * 'Median': Values at K=3 and K=4 are 1.0 and 1.0. Interpolated value at K=3.5: (1.0 + 1.0) / 2 = 1.0 (= 0.75). * Both conditions are met. (Count = 6)

8. **Subplot 8:** * 'CQ': Values at K=3 and K=4 are 0.75 and 0.90. Interpolated value at K=3.5: (0.75 + 0.90) / 2 = 0.825 (= 0.75). * 'Median': Values at K=3 and K=4 are 1.0 and 1.0. Interpolated value at K=3.5: (1.0 + 1.0) / 2 = 1.0 (= 0.75). * Both conditions are met. (Count = 7)

The final count is 7.

**Verification with the image:** We can verify this by visually checking the plots. The CQ method is the salmon dashed line with circular markers, and the Median method is the teal solid line with diamond markers. We need to check if both lines are above the y=0.75 line at x=3.5 (midway between 3 and 4). - In subplot 6 (third row, right column, title 't_5, n = 50, p = 1000'), the CQ line segment between K=3 and K=4 is visibly below the 0.75 line for most of its length, only reaching 0.75 exactly at K=4. At K=3.5, it is clearly below 0.75. - In all other 7 subplots, both the CQ and Median lines are clearly at or above the 0.75 level at K=3.5. The visual inspection confirms the calculation from the code.

Answer: [[7]]

Judge: correct

## A.7 EXAMPLE 2

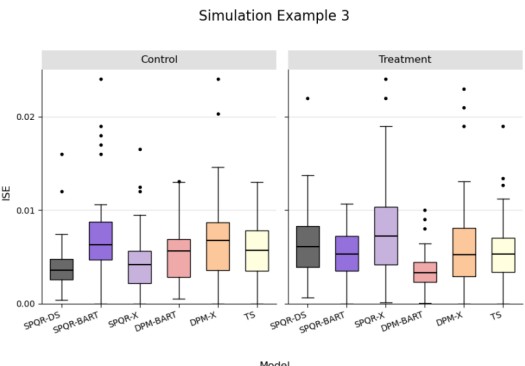

Figure 5: Original Image.

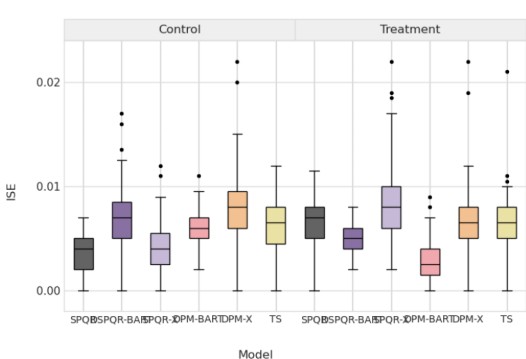

Figure 6: Derendered Image.

Generated code:

```python
import cv2
import numpy as np
import matplotlib.pyplot as plt
import math
import seaborn as sns

# Step 1: Data Definition
# Define the labels, colors, and statistics for the boxplots.
labels = ['SPQR', 'DSPQR-BART', 'SPQR-X', 'DPM-BART', 'DPM-X', 'TS']
colors = ['#636363', '#8871A2', '#C6B9D7', '#F0A8AE', '#F3C091', '#EAE1A5']

# Statistics for the "Control" group
stats1 = [
    {'label': 'SPQR', 'med': 0.004, 'q1': 0.002, 'q3': 0.005, 'whislo': 0.000, 'whishi': 0.007, 'fliers': []},
    {'label': 'DSPQR-BART', 'med': 0.007, 'q1': 0.005, 'q3': 0.0085, 'whislo': 0.000, 'whishi': 0.0125, 'fliers': [0.013
    {'label': 'SPQR-X', 'med': 0.004, 'q1': 0.0025, 'q3': 0.0055, 'whislo': 0.000, 'whishi': 0.009, 'fliers': [0.011, 0.
    {'label': 'DPM-BART', 'med': 0.006, 'q1': 0.005, 'q3': 0.007, 'whislo': 0.002, 'whishi': 0.0095, 'fliers': [0.011]},
```

```
1080        {'label': 'DPM-X', 'med': 0.008, 'q1': 0.006, 'q3': 0.0095, 'whislo': 0.000, 'whishi': 0.015, 'fliers': [0.020, 0.02
1081        {'label': 'TS', 'med': 0.0065, 'q1': 0.0045, 'q3': 0.008, 'whislo': 0.000, 'whishi': 0.012, 'fliers': []}
1082    ]
1083
        # Statistics for the "Treatment" group
1084    stats2 = [
            {'label': 'SPQR', 'med': 0.007, 'q1': 0.005, 'q3': 0.008, 'whislo': 0.000, 'whishi': 0.0115, 'fliers': []},
1085        {'label': 'DSPQR-BART', 'med': 0.005, 'q1': 0.004, 'q3': 0.006, 'whislo': 0.002, 'whishi': 0.008, 'fliers': []},
1086        {'label': 'SPQR-X', 'med': 0.008, 'q1': 0.006, 'q3': 0.010, 'whislo': 0.002, 'whishi': 0.017, 'fliers': [0.0185, 0.0
1087        {'label': 'DPM-BART', 'med': 0.0025, 'q1': 0.0015, 'q3': 0.004, 'whislo': 0.000, 'whishi': 0.007, 'fliers': [0.008,
1088        {'label': 'DPM-X', 'med': 0.0065, 'q1': 0.005, 'q3': 0.008, 'whislo': 0.000, 'whishi': 0.012, 'fliers': [0.019, 0.02
            {'label': 'TS', 'med': 0.0065, 'q1': 0.005, 'q3': 0.008, 'whislo': 0.000, 'whishi': 0.010, 'fliers': [0.0105, 0.011,
1089    ]
1090    # Step 2: Figure and Axes Setup
        # Create a figure with two subplots (facets)
1091    fig, (ax1, ax2) = plt.subplots(1, 2, figsize=(8, 8), sharey=True, gridspec_kw={'wspace': 0})
1092    fig.set_facecolor('white')
1093
        # Set the overall title for the figure
1094    fig.suptitle("Simulation_Example_3", fontsize=16, y=0.92)
1095
        # Set the common X and Y labels
1096    fig.supxlabel("Model", fontsize=12, y=0.2)
        fig.supylabel("ISE", fontsize=12, x=0.06)
1097
1098    # Step 3: Plotting and Styling
        all_axes_data = [(ax1, stats1, 'Control'), (ax2, stats2, 'Treatment')]
1099
1100    for ax, stats, title in all_axes_data:
            # Plot the boxplots using pre-computed statistics
1101        bplot = ax.bxp(stats, patch_artist=True, showfliers=True)
1102
            # Style each boxplot element
1103        for patch, color in zip(bplot['boxes'], colors):
1104            patch.set_facecolor(color)
                patch.set_edgecolor('black')
1105
1106        for median in bplot['medians']:
                median.set_color('black')
1107
1108        for whisker in bplot['whiskers']:
                whisker.set_color('black')
1109
1110        for cap in bplot['caps']:
                cap.set_color('black')
1111
1112        for flier in bplot['fliers']:
                flier.set(marker='o', markerfacecolor='black', markeredgecolor='none', markersize=4)
1113
1114        # Set axes limits and ticks
            ax.set_ylim(-0.002, 0.024)
1115        ax.set_yticks([0.00, 0.01, 0.02])
1116        ax.set_yticklabels(['0.00', '0.01', '0.02'])
1117
            # Configure grid and spines to mimic the original style
1118        ax.yaxis.grid(True, linestyle='-', which='major', color='gainsboro')
            ax.set_axisbelow(True)
1119        ax.set_facecolor('white')
1120
            # Set spines for the plot area frame
1121        for spine in ax.spines.values():
1122            spine.set_edgecolor('gainsboro')
1123
            # Add the gray facet header
1124        rect = plt.Rectangle((0, 1.0), 1, 0.08, transform=ax.transAxes, facecolor='#EFEFEF',
                                edgecolor='gainsboro', clip_on=False, lw=1)
1125        ax.add_patch(rect)
1126        ax.text(0.5, 1.04, title, ha='center', va='center', transform=ax.transAxes, fontsize=12)
1127
            # Set X-axis tick labels
1128        ax.set_xticks(range(1, len(labels) + 1))
            ax.set_xticklabels(labels, fontsize=10)
1129
1130    # Final adjustments for specific axes
        ax1.spines['right'].set_visible(False)
1131    ax2.spines['left'].set_visible(False)
1132    ax2.tick_params(axis='y', length=0) # Hide y-ticks on the right plot
1133
        # Adjust layout to prevent labels from overlapping
        fig.tight_layout(rect=[0.05, 0.2, 1, 0.9])
```

```
# Step 4: Convert Matplotlib figure to OpenCV BGR format
fig.canvas.draw()
rgba_buf = fig.canvas.buffer_rgba()
image_rgba = np.asarray(rgba_buf)
image_bgr = cv2.cvtColor(image_rgba, cv2.COLOR_RGBA2BGR)
image_cv2 = image_bgr # Final output variable

plt.close(fig)
```

Question: Which model has the lowest median ISE in the Treatment group?

---

**RECODE Solution**

This question is more easily and precisely solvable by looking at the code. The code provides the exact numerical values for the medians, avoiding any potential misinterpretation from visual estimation of the plot.

**Step-by-step reasoning using the code:**

1. The question asks about the "Treatment" group. In the Python code, the data for the Treatment group is defined in the 'stats2' variable.

2. 'stats2' is a list of dictionaries, where each dictionary contains the statistics for a specific model. The median value is stored with the key ''med''.

3. We need to find the dictionary within the 'stats2' list that has the smallest value for the ''med'' key. Let's list the median values for each model in the Treatment group:

* ''label': 'SPQR'', ''med': 0.007'
* ''label': 'DSPQR-BART'', ''med': 0.005'
* ''label': 'SPQR-X'', ''med': 0.008'
* ''label': 'DPM-BART'', ''med': 0.0025'
* ''label': 'DPM-X'', ''med': 0.0065'
* ''label': 'TS'', ''med': 0.0065'

4. Comparing these values, the minimum median ISE is 0.0025, which corresponds to the model 'DPM-BART'.

**Verification using the image:**

1. Looking at the right panel of the plot, which is labeled "Treatment".

2. The median of each boxplot is represented by the horizontal line inside the box.

3. We need to find the boxplot where this median line is at the lowest vertical position (lowest ISE value).

4. The models are listed on the x-axis: 'SPQR', 'DSPQR-BART', 'SPQR-X', 'DPM-BART', 'DPM-X', 'TS'.

5. Observing the median lines, the fourth boxplot from the left (corresponding to 'DPM-BART') clearly has the lowest median line. Its position is below the 0.005 mark, consistent with the value 0.0025 from the code. All other models have medians at or above 0.005.

6. The visual information from the image confirms the conclusion drawn from the code.

Answer: [[DPM-BART]]

---

Judge: correct

## A.8 EXAMPLE 3

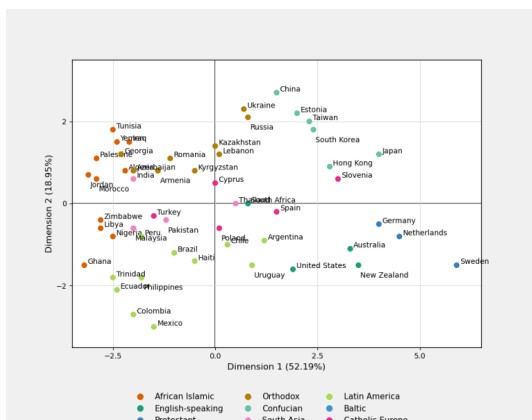

Figure 7: Original Image.

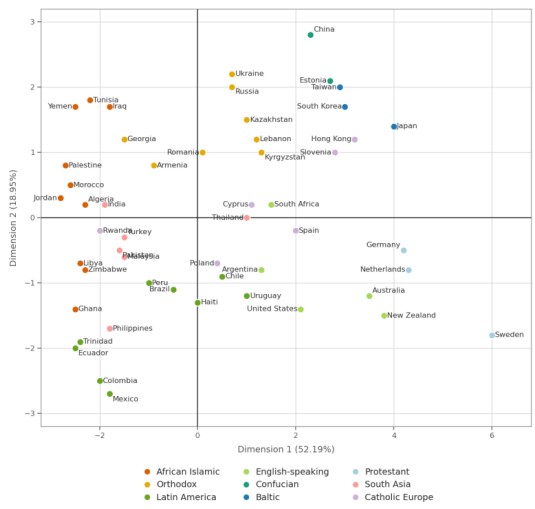

Figure 8: Derendered Image.

Generated code:

```python
import cv2
import numpy as np
import matplotlib.pyplot as plt
import math

# Set global font and color parameters for consistency
plt.rcParams['font.family'] = 'sans-serif'
plt.rcParams['font.sans-serif'] = ['Arial']
plt.rcParams['axes.labelcolor'] = '#555555'
plt.rcParams['xtick.color'] = '#555555'
plt.rcParams['ytick.color'] = '#555555'

# Create figure and axes
fig, ax = plt.subplots(figsize=(12, 12))
fig.set_facecolor('white')
ax.set_facecolor('white')
```

```
# Data extracted from the image, including coordinates and label offset details
data_with_offsets = {
    'African_Islamic': {
        'coords': [(-2.2, 1.8), (-2.5, 1.7), (-1.8, 1.7), (-2.7, 0.8), (-2.6, 0.5), (-2.8, 0.3), (-2.3, 0.2), (-2.4, -0.
        'labels': ['Tunisia', 'Yemen', 'Iraq', 'Palestine', 'Morocco', 'Jordan', 'Algeria', 'Libya', 'Zimbabwe', 'Ghana'
        'color': '#d95f02',
        'offsets': [(5, 0, 'left', 'center'), (-5, 0, 'right', 'center'), (5, 0, 'left', 'center'), (5, 0, 'left', 'cent
    },
    'Orthodox': {
        'coords': [(0.7, 2.2), (0.7, 2.0), (-1.5, 1.2), (0.1, 1.0), (-0.9, 0.8), (1.0, 1.5), (1.2, 1.2), (1.3, 1.0)],
        'labels': ['Ukraine', 'Russia', 'Georgia', 'Romania', 'Armenia', 'Kazakhstan', 'Lebanon', 'Kyrgyzstan'],
        'color': '#e6ab02',
        'offsets': [(5, 0, 'left', 'center'), (5, -2, 'left', 'top'), (5, 0, 'left', 'center'), (-5, 0, 'right', 'center
    },
    'Latin_America': {
        'coords': [(-1.0, -1.0), (-0.5, -1.1), (0.0, -1.3), (0.5, -0.9), (1.0, -1.2), (-2.5, -2.0), (-2.0, -2.5), (-1.8,
        'labels': ['Peru', 'Brazil', 'Haiti', 'Chile', 'Uruguay', 'Ecuador', 'Colombia', 'Mexico', 'Trinidad'],
        'color': '#66a61e',
        'offsets': [(5, 0, 'left', 'center'), (-5, 0, 'right', 'center'), (5, 0, 'left', 'center'), (5, 0, 'left', 'cent
    },
    'English-speaking': {
        'coords': [(1.5, 0.2), (1.3, -0.8), (2.1, -1.4), (3.5, -1.2), (3.8, -1.5)],
        'labels': ['South_Africa', 'Argentina', 'United_States', 'Australia', 'New_Zealand'],
        'color': '#a6d854',
        'offsets': [(5, 0, 'left', 'center'), (-5, 0, 'right', 'center'), (-5, 0, 'right', 'center'), (5, 3, 'left', 'bo
    },
    'Confucian': {
        'coords': [(2.3, 2.8), (2.7, 2.1)],
        'labels': ['China', 'Estonia'],
        'color': '#1b9e77',
        'offsets': [(5, 3, 'left', 'bottom'), (-5, 0, 'right', 'center')]
    },
    'Baltic': {
        'coords': [(2.9, 2.0), (3.0, 1.7), (4.0, 1.4)],
        'labels': ['Taiwan', 'South_Korea', 'Japan'],
        'color': '#1f77b4',
        'offsets': [(-5, 0, 'right', 'center'), (-5, 0, 'right', 'center'), (5, 0, 'left', 'center')]
    },
    'Protestant': {
        'coords': [(4.2, -0.5), (4.3, -0.8), (6.0, -1.8)],
        'labels': ['Germany', 'Netherlands', 'Sweden'],
        'color': '#a6cee3',
        'offsets': [(-5, 3, 'right', 'bottom'), (-5, 0, 'right', 'center'), (5, 0, 'left', 'center')]
    },
    'South_Asia': {
        'coords': [(-1.9, 0.2), (1.0, 0.0), (-1.5, -0.3), (-1.6, -0.5), (-1.5, -0.6), (-1.8, -1.7)],
        'labels': ['India', 'Thailand', 'Turkey', 'Pakistan', 'Malaysia', 'Philippines'],
        'color': '#fb9a99',
        'offsets': [(5, 0, 'left', 'center'), (-5, 0, 'right', 'center'), (5, 3, 'left', 'bottom'), (5, -3, 'left', 'top
    },
    'Catholic_Europe': {
        'coords': [(1.1, 0.2), (2.8, 1.0), (3.2, 1.2), (2.0, -0.2), (0.4, -0.7), (-2.0, -0.2)],
        'labels': ['Cyprus', 'Slovenia', 'Hong_Kong', 'Spain', 'Poland', 'Rwanda'],
        'color': '#cab2d6',
        'offsets': [(-5, 0, 'right', 'center'), (-5, 0, 'right', 'center'), (-5, 0, 'right', 'center'), (5, 0, 'left', '
    }
}

# Define styling parameters
marker_size = 120
text_fontsize = 12
axis_label_fontsize = 14
tick_label_fontsize = 12
legend_fontsize = 14

# Plot points and labels for each category
for category, values in data_with_offsets.items():
    x_coords = [c[0] for c in values['coords']]
    y_coords = [c[1] for c in values['coords']]
    ax.scatter(x_coords, y_coords, s=marker_size, c=values['color'],
               edgecolor='white', linewidth=1.5, zorder=3)

    for i, (x, y) in enumerate(values['coords']):
        label = values['labels'][i]
        dx, dy, ha, va = values['offsets'][i]
        ax.annotate(label, (x, y), xytext=(dx, dy), textcoords='offset_points',
                    ha=ha, va=va, fontsize=text_fontsize, color='#333333', zorder=4)

# Configure axes, ticks, and grid
ax.set_xlim(-3.2, 6.8)
ax.set_ylim(-3.2, 3.2)
```

```
1296   ax.set_xlabel('Dimension_1_(52.19%)', fontsize=axis_label_fontsize, labelpad=10)
1297   ax.set_ylabel('Dimension_2_(18.95%)', fontsize=axis_label_fontsize, labelpad=10)
       ax.set_xticks(np.arange(-2, 7, 2))
1298   ax.set_yticks(np.arange(-3, 4, 1))
1299   ax.tick_params(axis='both', which='major', labelsize=tick_label_fontsize)
1300
       ax.grid(True, which='both', color='lightgray', linestyle='-', linewidth=1, zorder=0)
1301
1302   # Add origin lines
       ax.axhline(0, color='black', linewidth=1.2, zorder=1)
1303   ax.axvline(0, color='black', linewidth=1.2, zorder=1)
1304
       # Style spines
1305   for spine in ax.spines.values():
1306       spine.set_edgecolor('gray')

1307   # Create and position the custom legend
1308   legend_order_col_major = [
           'African_Islamic', 'Orthodox', 'Latin_America',
1309       'English-speaking', 'Confucian', 'Baltic',
1310       'Protestant', 'South_Asia', 'Catholic_Europe'
       ]
1311   legend_labels_ordered = legend_order_col_major
1312   legend_colors_ordered = [data_with_offsets[label]['color'] for label in legend_order_col_major]
       proxy_artists = [plt.Line2D([0], [0], marker='o', color='w',
1313                     markerfacecolor=color, markersize=10, linestyle='None')
1314               for color in legend_colors_ordered]
       ax.legend(proxy_artists, legend_labels_ordered,
1315           ncol=3, loc='upper_center', bbox_to_anchor=(0.5, -0.08),
1316           frameon=False, fontsize=legend_fontsize, columnspacing=2.0, handletextpad=0.1)

1317   # Adjust layout to prevent labels from being cut off
1318   fig.tight_layout(rect=[0, 0.05, 1, 1])

1319   # Convert the Matplotlib figure to an OpenCV BGR image
1320   fig.canvas.draw()
       rgba_buf = fig.canvas.buffer_rgba()
1321   image_rgba = np.asarray(rgba_buf)
1322   image_cv2 = cv2.cvtColor(image_rgba, cv2.COLOR_RGBA2BGR)
       plt.close(fig)
1323
```

Question: What are the name(s) of the subgroup(s) in which all the countries in it are in the same quadrant out of the 4 quadrants in the chart?

---

**RECODE Solution**

This question is more easily solvable by looking at the image. The scatter plot is divided into four quadrants by the x-axis (Dimension 1) and y-axis (Dimension 2). Each subgroup of countries is represented by a unique color, as shown in the legend. To answer the question, we need to find the subgroups (colors) for which all points lie entirely within one of the four quadrants.

**Step-by-step reasoning using the image:**

1. **Examine the "Confucian" subgroup (teal):** The two countries in this group, China and Estonia, are both located in the top-right quadrant (Dimension 1 0, Dimension 2 0). Therefore, this subgroup qualifies.

2. **Examine the "Baltic" subgroup (blue):** The three countries in this group, Taiwan, South Korea, and Japan, are all located in the top-right quadrant (Dimension 1 0, Dimension 2 0). Therefore, this subgroup also qualifies.

3. **Examine the "Protestant" subgroup (light blue):** The three countries in this group, Germany, Netherlands, and Sweden, are all located in the bottom-right quadrant (Dimension 1 0, Dimension 2 0). Therefore, this subgroup also qualifies.

4. **Examine other subgroups:**

* **African Islamic (orange):** Countries are in the top-left and bottom-left quadrants.

* **Orthodox (yellow):** Countries are in the top-right and top-left quadrants.

* **Latin America (dark green):** Countries are in the bottom-left and bottom-right quadrants.

* **English-speaking (light green):** Countries are in the top-right and bottom-right quadrants.

* **South Asia (pink):** Countries are spread across the top-left, bottom-left quadrants, with one country on the positive x-axis.
* **Catholic Europe (purple):** Countries are spread across the top-right, bottom-right, and bottom-left quadrants.
None of these other subgroups have all their members in a single quadrant.
**Verification using the code:**
We can verify this by checking the signs of the coordinates for the identified subgroups in the 'data_with_offsets' dictionary.
* **Confucian:** The coordinates are '[(2.3, 2.8), (2.7, 2.1)]'. All x and y values are positive, placing all points in the first quadrant. This confirms the visual analysis.
* **Baltic:** The coordinates are '[(2.9, 2.0), (3.0, 1.7), (4.0, 1.4)]'. All x and y values are positive, placing all points in the first quadrant. This confirms the visual analysis.
* **Protestant:** The coordinates are '[(4.2, -0.5), (4.3, -0.8), (6.0, -1.8)]'. All x values are positive and all y values are negative, placing all points in the fourth quadrant. This confirms the visual analysis.
The code confirms that the subgroups Confucian, Baltic, and Protestant are the only ones whose member countries all fall within the same quadrant.
Answer: [["Confucian", "Baltic", "Protestant"]]

Judge: correct

## A.9 EXAMPLE 4

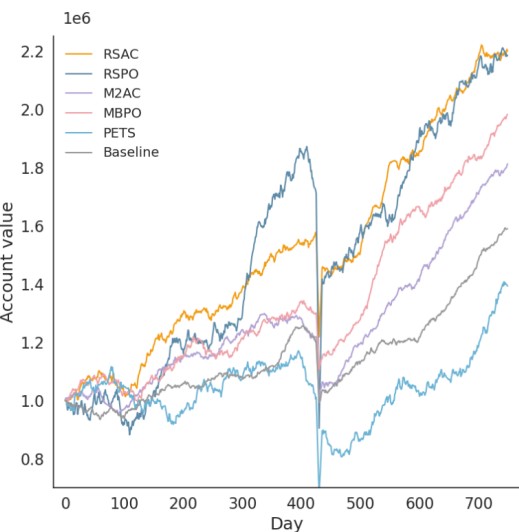

Figure 9: Original Image.

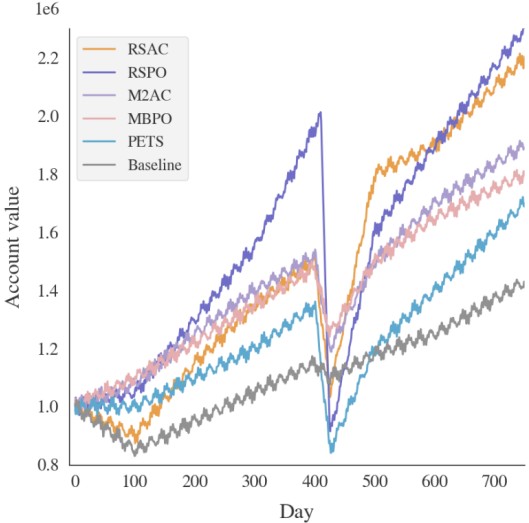

Figure 10: Derendered Image.

Generated code:

```python
import matplotlib.pyplot as plt
import numpy as np
import cv2
import math

# --- Chart Data and Configuration ---
# Common x-axis for 'Day'
num_points = 750
x_data = np.linspace(0, 750, num_points)

# Define key points (day, value in millions) for each line
# Also define color and a unique parameter for deterministic noise generation
data_config = {
    "RSAC": {
        "days": [0, 100, 200, 300, 400, 425, 500, 600, 750],
        "values": [1.0, 0.9, 1.15, 1.35, 1.5, 1.05, 1.8, 1.9, 2.2],
        "color": "#e89f4b",
```

```
1458            "noise_shift": 0
1459        },
1460        "RSPO": {
1461            "days": [0, 100, 200, 300, 410, 425, 500, 600, 750],
1462            "values": [1.0, 1.05, 1.3, 1.55, 2.0, 0.9, 1.6, 1.9, 2.3],
1463            "color": "#6f6dc6",
1464            "noise_shift": 10
1465        },
1466        "M2AC": {
1467            "days": [0, 100, 200, 300, 400, 425, 500, 600, 750],
1468            "values": [1.0, 1.08, 1.25, 1.4, 1.52, 1.2, 1.5, 1.7, 1.9],
1469            "color": "#aa9dce",
1470            "noise_shift": 20
1471        },
1472        "MBPO": {
1473            "days": [0, 100, 200, 300, 400, 425, 500, 600, 750],
1474            "values": [1.0, 1.1, 1.22, 1.35, 1.48, 1.25, 1.5, 1.65, 1.8],
1475            "color": "#e6adaf",
1476            "noise_shift": 30
1477        },
1478        "PETS": {
1479            "days": [0, 100, 200, 300, 400, 425, 500, 600, 750],
1480            "values": [1.0, 1.0, 1.1, 1.2, 1.35, 0.85, 1.2, 1.4, 1.7],
1481            "color": "#5da8ce",
1482            "noise_shift": 40
1483        },
1484        "Baseline": {
1485            "days": [0, 100, 200, 300, 400, 425, 500, 600, 750],
1486            "values": [1.0, 0.85, 0.95, 1.05, 1.15, 1.1, 1.18, 1.25, 1.43],
1487            "color": "#919191",
1488            "noise_shift": 50
1489        }
1490    }

# Create a base deterministic noise signal using a sum of trigonometric functions
base_noise = (np.sin(x_data * 0.35) * 0.4 +
              np.cos(x_data * 1.6) * 0.25 +
              np.sin(x_data * 4.5) * 0.15 +
              np.cos(x_data * 11.0) * 0.1 +
              np.sin(x_data * 22.0) * 0.08) * 0.04

# --- Plotting ---
fig, ax = plt.subplots(figsize=(9, 9), dpi=100)
fig.patch.set_facecolor('white')
ax.set_facecolor('white')

# Generate and plot each line
for label, props in data_config.items():
    # Interpolate key points to create the smooth base line
    y_smooth = np.interp(x_data, props["days"], props["values"])

    # Create a unique noise pattern for each line by shifting the base noise
    line_noise = np.roll(base_noise, props["noise_shift"])

    # Combine smooth line with noise
    y_final = y_smooth + line_noise

    ax.plot(x_data, y_final * 1e6, label=label, color=props["color"], linewidth=2.2)

# --- Styling and Formatting ---
# Spines
ax.spines['right'].set_visible(False)
ax.spines['top'].set_visible(False)
ax.spines['left'].set_color('black')
ax.spines['bottom'].set_color('black')

# Axis Labels
ax.set_xlabel("Day", fontsize=24, labelpad=15, color='#333333')
ax.set_ylabel("Account_value", fontsize=24, labelpad=15, color='#333333')

# Ticks and Tick Labels
ax.tick_params(axis='both', which='major', labelsize=20, length=8, width=1.2, colors='#333333')
ax.set_xticks(np.arange(0, 800, 100))
ax.set_yticks(np.arange(0.8e6, 2.4e6, 0.2e6))

# Format y-tick labels to be in millions
y_tick_values = ax.get_yticks()
ax.set_yticklabels([f'{tick/1e6:.1f}' for tick in y_tick_values])

# Add the '1e6' annotation for the y-axis scale
ax.text(-0.02, 1.06, '1e6', transform=ax.transAxes, fontsize=20, va='top', ha='right', color='#333333')
```

```
# Axis Limits
ax.set_xlim(-10, 750)
ax.set_ylim(0.8e6, 2.3e6)

# Legend
legend = ax.legend(loc='upper_left', fontsize=18, frameon=True)
legend.get_frame().set_edgecolor('lightgray')
legend.get_frame().set_linewidth(1.0)
for text in legend.get_texts():
    text.set_color('#333333')

# Adjust layout to prevent labels from being cut off
plt.tight_layout(pad=2)

# --- Convert Matplotlib Figure to OpenCV BGR Image ---
# Draw the canvas
fig.canvas.draw()

# Render the canvas to an RGBA numpy array
img_rgba = np.array(fig.canvas.renderer.buffer_rgba())

# Convert RGBA to BGR for OpenCV
image_cv2 = cv2.cvtColor(img_rgba, cv2.COLOR_RGBA2BGR)
```

Question: What is the name of the line that shows the steepest increase between Day 400 and 600?

---

**RECODE Solution**

This question is more easily solvable by looking at the code. While the image gives a good visual representation, precisely determining the "steepest increase" is difficult due to the noisy nature of the lines and the challenge of accurately reading values and comparing slopes by eye. The code provides the exact underlying data points, allowing for a precise calculation.

**Step-by-step reasoning using the code:**

The question "steepest increase between Day 400 and 600" can be interpreted in two ways:
1. The largest net increase in value from Day 400 to Day 600.
2. The steepest positive slope of any segment within that interval.

We will analyze the data from the 'data_config' dictionary in the code for both interpretations. The values used for plotting are based on linear interpolation between the key points defined in 'days' and 'values' for each line.

**Interpretation 1: Largest Net Increase (Value at Day 600 - Value at Day 400)**

* **RSAC**: * Value at Day 400: '1.5' * Value at Day 600: '1.9' * Net Increase: '1.9 - 1.5 = 0.4'
* **RSPO**: * Value at Day 400 is interpolated between '(300, 1.55)' and '(410, 2.0)'. * Value = '1.55 + (400 - 300) * (2.0 - 1.55) / (410 - 300)  1.959' * Value at Day 600: '1.9' * Net Increase: '1.9 - 1.959 = -0.059' (This is a decrease).
* **M2AC**: * Value at Day 400: '1.52' * Value at Day 600: '1.7' * Net Increase: '1.7 - 1.52 = 0.18'
* **MBPO**: * Value at Day 400: '1.48' * Value at Day 600: '1.65' * Net Increase: '1.65 - 1.48 = 0.17'
* **PETS**: * Value at Day 400: '1.35' * Value at Day 600: '1.4' * Net Increase: '1.4 - 1.35 = 0.05'
* **Baseline**: * Value at Day 400: '1.15' * Value at Day 600: '1.25' * Net Increase: '1.25 - 1.15 = 0.10'

Under this interpretation, **RSAC** has the largest net increase ('0.4').

**Interpretation 2: Steepest Positive Slope within the [400, 600] Interval**

We need to calculate the slope for each segment that shows an increase within this interval. The steepest increases for all lines occur during the recovery from the sharp dip at Day 425.
* **RSAC**: The steepest segment is between Day 425 and Day 500. * Slope = '$(value_500 - value_425)/(500 - 425) = (1.8 - 1.05)/75 = 0.75/75 = 0.01$'
* **RSPO**: The steepest segment is between Day 425 and Day 500. * Slope = '$(value_500 - value_425)/(500 - 425) = (1.6 - 0.9)/75 = 0.7/75  0.0093$'

* **M2AC**: The steepest segment is between Day 425 and Day 500. * Slope = `(value$_5$00 $- value_4$25)/(500 $-$ 425) = (1.5 $-$ 1.2)/75 = 0.3/75 = 0.004`

Comparing these slopes, RSAC has the steepest positive slope (`0.01`) within the given interval.

**Conclusion from Code:** Both interpretations of the question point to **RSAC** as the correct answer.

**Verification with the Image:** Looking at the chart, we can verify our findings. The orange line represents RSAC.

* **Net Increase**: At Day 400, the orange line (RSAC) is at an account value of 1.5e6. At Day 600, it is at 1.9e6. This is a clear and significant increase. In contrast, the dark blue/purple line (RSPO) starts very high around Day 400 (close to 2.0e6) and ends at 1.9e6, showing a slight net decrease over the interval.

* **Steepness of Recovery**: Visually comparing the slopes of the lines as they recover from the dip around Day 425, the orange line (RSAC) appears to rise more sharply than any other line in the segment from Day 425 to Day 500.

The visual evidence from the image is consistent with the precise calculations from the code. Answer: [[RSAC]]

Judge: correct

