# OpenReview forum: "RECODE: Reasoning Through Code Generation for Visual Question Answering"
_ICLR.cc/2026/Conference — Submitted to ICLR 2026_

### Official Review · Reviewer_kcXj · 2025-10-16

**Soundness:** 2
**Presentation:** 1
**Contribution:** 2
**Rating:** 2
**Confidence:** 5

**Summary:**

The paper introduces RECODE, a multimodal agentic framework for visual question answering that approaches visual reasoning on structured images  by derendering these inputs into executable code. The RECODE agent generates multiple candidate programs to represent the generative logic of the input image, employs a critic to select and iteratively refine the most faithful code, and then leverages both code and visual modalities for precise, verifiable reasoning. Comprehensive results across benchmarks show that RECODE significantly improves accuracy over both direct pixel-based VQA and recent tool-augmented agents.

**Strengths:**

1.  The integration of a closed self-improving loop (generation, critic selection, iterative code refinement) offers a robust agentic architecture. As illustrated in Figure 1, the pipeline systematically closes the gap between generated and true underlying code, enabling higher accuracy and verifiability.
2. The paper convincingly reframes structured visual reasoning as a code generation/derendering problem, moving from ambiguous perception towards a verifiable, interpretable, symbolic representation. This is a meaningful shift in thinking for VQA on charts and diagrams.

**Weaknesses:**

1. Much of the empirical validation especially for the initial proof-of-concept and ablation studies around derendering operates on data where ground-truth generative code can be plausibly extracted or synthesized. While the pipeline is motivated as general, there’s insufficient evidence that it scales robustly to complex real-world scientific visuals with lossy, noisy rendering, rasterization artifacts, or hand-drawn figures without programmatic sources. The argument that RECODE generalizes “beyond clean benchmark imagery” is not substantiated in results or examples.
2. The entire experiment was only conducted on the Gemini2.5 pro, and the benchmark measurements were relatively few, lacking generalizable discussions.
3. The ablation and main experiments do not address settings with highly cluttered or multi-pane charts, severe occlusion, transparent overlays, or dense text. There’s only limited qualitative evidence (Appendix A.6) that RECODE remains robust as the generative complexity increases. Without experiments on images pushing these boundaries, claims of general applicability are unproven.
4. Some code snippets in the appendix (A.6, A.7) and their outputs suffer from typographical or formatting errors (e.g., code blocks with syntax issues or unmatched parentheses), which could hinder replication or adaptation. This led to a very messy layout of the entire paper.
5. Some recent advances on multimodal visual code reasoning (e.g., SWE-bench Multimodal, CodeV, SceneCraft, 3D-GPT) that tackle similar cross-modal code generation or VQA topics are not cited or discussed, limiting the contextualization of the contribution.
6. While some error cases are mentioned (Section 4.3, discussing multi-line color ambiguities), the paper lacks a candid characterization of persistent failure cases. For example, illustrations where OCR is unreliable, or compositional logic in diagrams exceeds single-pass code generation capacity—even after iterative refinement.
7. Details regarding the number of refinement rounds and candidate generations per round are sparsely justified (“two rounds, five candidates per round” is asserted as default). It is not clear how sensitive major results are to these settings, or whether performance plateaus or collapses with more iterations. There is insufficient reporting on convergence/stability.

**Questions:**

1. Can the authors provide quantitative or qualitative results on inputs with high visual noise, hand-drawn charts, or non-programmatically-generated figures to better test generalization?
2. How does RECODE perform when the ground-truth code is deeply non-deterministic, or uses uncommon libraries/functions not in the agent’s training distribution? Is there a clear drop-off in accuracy, or does refinement reliably recover?
3. Could the authors elaborate on what (if any) failure cases persist after two or more refinement rounds? Is there a systematic way to detect "irrecoverable" mismatches, such as OCR failures or data occlusion, beyond visual similarity in the critic?
4. What are the sensitivity and convergence behaviors as the number of candidates and refinement rounds is increased (beyond the default values used in the paper)?
5. Do the authors have strategies or insights on expanding to settings with multiple heterogeneous subplots or compositional charts, where subfigure logic or relative positioning becomes essential to the answer?
6. Given that the method relies on visual code synthesis, what are the possible risks regarding privacy, security, or copyright, especially for proprietary or sensitive scientific visuals?
7. Can the author conduct tests and verifications on more benchmarks or more base models to verify the generalization of the method？

**Details Of Ethics Concerns:**

The core method could be used to reverse-engineer proprietary, copyrighted, or watermark-protected images into reusable code, possibly violating data licensing or intellectual property. This should be discussed by the authors in the context of responsible research and deployment, especially since code reverse engineering may have implications on research integrity and legal compliance. While no immediate evidence indicates intent for misuse or direct harm, this is a broader risk of automation in this space.

---

### Official Review · Reviewer_EbgW · 2025-10-31

**Soundness:** 3
**Presentation:** 4
**Contribution:** 3
**Rating:** 6
**Confidence:** 3

**Summary:**

For challenging visual-related questions such as those for charts or diagrams, the authors propose RECODE to generate executable codes that can reproduce the input images and use the generated code as a symbolic representation to help answering the visual-related questions. They iteratively refine the generated code by comparing the original image and the image generated by the code, The authors select the critic function as pixel-based MSE carefully to choose the best candidate code in each round. The experimental results are positive by showing RECODE surpasses open-sourced and even closed-sourced MLLMs in several chart/diagram VQA benchmarks.

**Strengths:**

1. The motivation is explained clearly. The authors use gemini2.5 pro for testing the accuracies of QA under three conditions: (1) given images (2) given GT code and (3) given both images and the GT code. Their preliminary results show the value of derendering the images into code. The code can help the visual reasoning.

2. The authors experiment on different benchmarks and the positive results are consistent.

3. Each choice has ablation studies. The presentation is very clean and readable.

**Weaknesses:**

1. The authors provide ablation studies about adding OCR in the code generation process, which is reasonable. It would be interesting by adding a baseline of using OCR results + image only as inputs.

2. Because of the refinement process, I think the cost time will be more expensive. The authors should provide some details about it.

3. In the table 4, the authors try refinement round from 0 to 2. If the number of round becomes larger, will the performances be influenced and what will be the trend? Also, have the authors tried to change the base model to see if the best number of round will be changed as well?

**Questions:**

Please refine the Figure 2, some of the text are not clear enough.

typo:
Line 248: LLM -> MLLM? I guess the "LLM" here indicates Gemini Pro 2.5?

---

### Official Review · Reviewer_v3Nj · 2025-11-01

**Soundness:** 2
**Presentation:** 3
**Contribution:** 2
**Rating:** 4
**Confidence:** 4

**Summary:**

The paper introduces RECODE, a framework that tackles visual question answering by converting charts and diagrams into executable code. It generates multiple code candidates to reproduce the input image, selects the best one using a critic, and iteratively refines it for higher fidelity. Once the code accurately represents the visual, the model reasons over both the image and code to answer questions. Experiments on CharXiv, ChartQA, and Geometry3K show improvements over baselines.

**Strengths:**

-	The paper is clearly written, and the experiments on several benchmarks make it straightforward to see how the method helps to chart-related benchmarks.
-	The idea of representing visuals through executable code is interesting and reasonable.

**Weaknesses:**

-	The experiments rely only on Gemini-2.5 Pro, which makes it hard to tell whether the proposed pipeline generalizes across different MLLMs. I would suggest adding results with at least one open-source model and maybe GPT-5 for comparison.
-	The evaluated benchmarks are relatively narrow, focusing mainly on chart and geometry datasets. It would be better to include more diverse or harder visual reasoning benchmarks (e.g., Humaneval-V [1]) to test broader applicability.
-	I think the paper could analyze more failure cases. Right now, it mostly shows successful examples, but understanding where the derendering or refinement fails would make the approach clearer and help others reproduce or improve it.
-	The computational cost of multi-candidate generation and iterative refinement isn’t discussed in detail. It would be useful to report runtime or cost analysis to understand the trade-off between accuracy and efficiency.



[1] https://arxiv.org/pdf/2410.12381

**Questions:**

Please see weaknesses.

---

### Meta-Review · Area_Chair_An2Y · 2026-01-04

**Summary:**

This paper proposes RECODE, an agentic framework that performs visual question answering on structured visuals by derendering images into executable code, iteratively refining the code via a critic, and reasoning over both the code and the image. Reviewers generally found the core idea of grounding visual reasoning in executable code interesting and potentially impactful, and acknowledged the clarity of presentation and consistent gains on several chart/diagram benchmarks.

However, reviewers raised substantial concerns regarding the limited empirical scope, lack of generalization evidence, exclusive reliance on a single closed-source MLLM, insufficient analysis of failure cases and computational cost, and missing discussion of ethical and legal implications of reverse-engineering visuals into code. Importantly, the authors did not provide a rebuttal, leaving all major concerns unaddressed.

**Reviewer Concerns:**

The authors did not submit a rebuttal, and therefore none of the reviewers’ concerns were addressed.

**Reviewer Scores:**

The reviewer scores would likely remain unchanged.

---

### Decision · Program_Chairs · 2026-01-26

Reject